# Proton tautomerism for strong polarization switching

Sachio Horiuchi[1], Kensuke Kobayashi[2], Reiji Kumai[2] & Shoji Ishibashi[3]

Ferroelectrics based on proton tautomerism are promising in low-field and above-room-temperature operations. Here seven organic ferroelectric crystals are examined to search for efficient switching of strong spontaneous polarization on proton tautomerism. Solution-grown crystals exhibit strong pinning of ferroelectric domain walls, but excellent switching performance is awakened by depinning domain walls under thermal annealing and/or repetitive bipolar pulses with a high voltage. Compared with ferroelectric polymers such as polyvinylidefluoride, the optimized polarizations are comparable or stronger in magnitude whereas the coercive fields are two orders of magnitude weaker. The polarization of croconic acid, in particular, breaks its own record for organic systems in increasing from 21 to $30\,\mu C\,cm^{-2}$ and now exceeds those of some commercial ferroelectric materials such as $SrBi_2Ta_2O_9$ and $BaTiO_3$. Optimization reduces the discrepancy of the spontaneous polarization with the results of the first-principles calculations to less than 15%. The cooperative roles of proton transfer and $\pi$-bond switching are discussed by employing the point-charge model and hydrogen-bond geometry.

[1] Flexible Electronics Research Center (FLEC), National Institute of Advanced Industrial Science and Technology (AIST), Tsukuba 305-8565, Japan. [2] Condensed Matter Research Center (CMRC) and Photon Factory, Institute of Materials Structure Science, High Energy Accelerator Research Organization (KEK), Tsukuba 305-0801, Japan. [3] Research Center for Computational Design of Advanced Functional Materials (CD-FMat), National Institute of Advanced Industrial Science and Technology (AIST), Tsukuba 305-8568, Japan. Correspondence and requests for materials should be addressed to S.H. (email: s-horiuchi@aist.go.jp).

Ferroelectrics are electrically polar substances in which the direction of spontaneous polarization is reversibly switchable under the influence of an external electric field. Owing to the strong polarization and high Curie point, ferroelectric oxides have gained prominence for many practical applications in information technology, such as non-volatile memory, capacitors, sensors, actuators, ultrasonic devices, and nonlinear optics applications[1–4].

Organic systems are free of both toxic and rare elements, well suited to highly productive printing (low-temperature solution process) into sheet devices, and are expected to be advantageous for emerging applications with cheap, disposal, flexible, wearable and/or implantable characteristics[5–8]. Organic ferroelectrics have found some piezoelectric applications such as sensors and acoustic devices, although their available compounds have been limited to polyvinylidefluoride (PVDF, $(CH_2CF_2)_n$) variants. In contrast, low-voltage-memory operation has long been a challenging issue for polymer ferroelectrics because of the high switching field exceeding several hundred kilovolts per centimetre. Recently, some small organic molecules have appeared as future potential alternatives; molecules in which proton transfer within the hydrogen bonds can invert the crystal polarity under much lower switching fields typically ranging from one to several tens of kilovolts per centimetre[9,10]. As a consequence, few-volt switching has been achieved even on printed micrometre-thick single-crystal films of 2-methylbenzimidazole (MBI)[11].

Among ferroelectric small molecules, croconic acid (CRCA)[12] has materialized the strongest polarization through a cooperative proton tautomerism mechanism[13–15] (called prototropy, abbreviated herein as PTM), which relocates a proton through the hydrogen bond and simultaneously interchanges the locations of a single bond and adjacent double bond (Fig. 1). This discovery was followed by the development of PTM ferroelectrics using the β-diketone enol O=C-C=C-OH moieties (that is, 2-phenylmalondialdehyde (PhMDA) and 3-hydroxyphenalenone (HPLN))[16], carboxylic acid O=C-OH moiety (that is, cyclobutene-1,2-dicarboxylic acid (CBDC))[16], and heterocyclic -N=C-NH- moieties (that is, MBI and 5,6-dichloro-2-methylbenzimidazole (DC-MBI))[17]. All these PTM ferroelectric crystals construct extended chains of intermolecular resonance-assisted hydrogen bonds (RAHB)[18], which are strengthened by the interplay with the conjugated π-bond system. The findings of ferroelectricity support the concept proposed by Haddon, Stillinger, and Carter[19] in 1982 that the hydrogen-bonded PTM system would be one of the representative models for memory functional molecular devices. In addition, the large polarization with significant electronic contribution has been closely related to the observations of very strong second-order nonlinear optical effects of CRCA such as second harmonic generation and terahertz radiation due to optical rectification[20,21].

Despite having excellent prospects, many organic ferroelectric crystals including those based on PTM have often exhibited some ambiguity in actual performance; polarization-electric field (P−E) hysteresis loops are often ill-shapen, and the much-lower-than-expected remanent polarization values depend strongly on the pristine crystals[22]. Recently, the investigation of the domain-property relationship by Kagawa et al.[23] has provided useful clues for the refinement of performance. In general, ferroelectric crystals adopt a multidomain structure in which ferroelectric domain walls (DWs) separate differently polarized sections (domains) and their sweeping motion changes the bulk polarization. Kagawa et al.[23] visualized the domain structures of a solution-grown crystal of acid-base alternating supramolecules by piezoresponse force microscopy (PFM) and demonstrated that imperfect switching arises from charged DWs being less mobile against the external field than neutral DWs. Herein, 'neutral' or 'charged' denotes the presence or absence of bound charges on DWs and depends on whether the relative angle between the domain-wall plane and the polarization vector **P** in the domains is parallel or antiparallel[23]. To avoid a huge depolarization field on the DWs, dense bound charges must be almost completely compensated for by mobile charges and/or immobile charged defects. It should be noted that these pinning sites (that is, charged DWs) are likely to be embedded simultaneously with the crystal growth at room temperature but can be eliminated by reconstructing the domain structures once after heating beyond the Curie point. Removal of charged DWs and compensating trapped charges turns out to be one strategy for efficient switching of the polarization in above-room-temperature ferroelectric supramolecular crystals.

Similar DW pinning is expected for solution-grown PTM ferroelectric crystals, because their Curie point is above room temperature or even beyond the thermal stability limit of the solid state. Sotome et al.[21] imaged changes in the domain structures of CRCA crystals under various electric fields by measuring the emission of terahertz radiation and found that charged DWs were always almost fixed in space and prevent bulk switching of the polarization. Previous P−E hysteresis measurements of CRCA[12] exploited both thermal annealing and repetitive switching for depinning DWs, although poor reproducibility of the best polarization ($21 \mu C \, cm^{-2}$) suggested that these procedures are still insufficiently optimized. Similarly, the P−E hysteresis loops of a CBDC crystal exhibited a polarization switching that was small in comparison with the theoretical results (2.9 versus $6.6 \mu C \, cm^{-2}$ along the c-direction)[16].

Here we revise the P−E hysteresis properties of solution-grown PTM ferroelectric crystals by finding effective optimization procedures. The optimized remanent polarizations ($P_r$) were evaluated in comparison with the results of first-principles calculations. We considered the cooperative roles of proton transfer and π-bond switching using the ion-displacive (point-charge) picture and hydrogen-bond geometry. Seven PTM ferroelectric compounds including a new compound allow for a systematic understanding of polari-

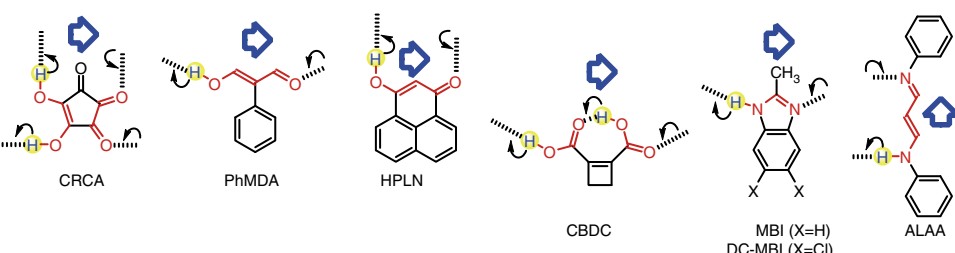

**Figure 1 | Chemical structural mechanisms of ferroelectric proton tautomerism.** The crystal polarity (open arrows) can be inverted by proton transfer (round arrow) over a hydrogen bond (thick broken line) and simultaneous interconversion of the single and double bonds, which are specified in red.

zation in search of design principles for high-performance switching.

## Results

**Materials and structural assessment**. Our previous studies have discovered six PTM ferroelectrics: CRCA, PhMDA, HPLN, CBDC, MBI and DC-MBI[12,16,17]. All these crystal structures comprise extended one- or two-dimensional networks through intermolecular hydrogen bonds; O-H···O bonds for the former four compounds and N-H···N bonds for the latter two. Their crystal symmetry and three-dimensional molecular packing are revisited in Supplementary Table 1 and Supplementary Figs 1–5 along with the crystal polarities. As shown schematically in Fig. 1, the polarity of the hydrogen-bonded networks is switchable through cooperative proton transfer and concomitant interchange of the double- and single-bond locations (the interconverting bonds and atoms involved are specified in red). In addition, the crystallographic requirement for ferroelectricity is a hidden pseudo-symmetry, which would survive as a paraelectric configuration.

Here, we also report a N-H···N bonded compound as the seventh PTM ferroelectric: 3-anilinoacrolein anil (ALAA)[24], which was encountered in the Cambridge Structural Database (CSD) (Refcode: ANPHPR). Because this entry does not include the atomic coordinates for the hydrogen atom, we have re-examined the crystal structure at room temperature ($T = 295$ K). The crystal structure belongs to the orthorhombic system with the polar space group $Iba2$ (#45) in agreement with the report. The crystal polarity is parallel to the crystal $c$-axis, which is along the direction that the hydrogen bonds construct an infinite chain of twists and turns. Without the N-H proton and $\pi$-bond alternation, the molecule can restore the pseudo-twofold rotation symmetry and occupy the $C_2$-site so as to constitute the centric (hypothetical) paraelectric structure: the space group $Ibca$ (#73). Although the hydrogen-bonded chain acquires a dipole switchable with the proton location, its polarization should mostly arise from the switchable $\pi$-bond dipole because all the protons travel in a direction normal to the polar axis. The orthogonality of the chain dipole and the proton's path is similar to the case of the HPLN crystal (Fig. 1 and Supplementary Fig. 6).

The possibility of 3-hydroxydibenzo[a,c]tropone (DBT, Supplementary Fig. 7a) exhibiting ferroelectricity was previously conjectured solely on the basis of earlier structural analysis of the polar crystal symmetry (monoclinic; space group $Cc$)[25]. In this investigation, DBT was synthesized according to the literature[26] and crystallized from solution. The $P - E$ hysteresis experiments unexpectedly revealed a signature of antiferroelectricity along the O-H···O bonded zigzag chain parallel to the crystal $c$-direction (Supplementary Figs 7 and 8). Through careful structural reassessments using a synchrotron X-ray source, antiferroelectric behaviour was attributed to the symmetry reduction from a $C$-centred to primitive lattice having antiparallel chain polarity arrangements. For details of the structural and electric characterizations, Supplementary Figs 7 and 8 and Supplementary Discussion.

**Theoretical polarization versus previous experiments**. According to the current dielectrics theory of solids, the evaluation of macroscopic polarization through the Berry phase formalism demands precise knowledge of the electronic structures in the crystal form[27,28]. Therefore, for all seven ferroelectrics, first-principles electronic structure calculations were performed to evaluate the spontaneous polarization $\mathbf{P}^{cal}$. For the target ferroelectric structure ($\lambda = 1$), the atomic coordinates of all the non-hydrogen atoms were obtained from

X-ray diffraction data. For the hydrogen atoms, the core locations were computationally relaxed so as to minimize the total energy, because the X-ray diffraction merely locates the corresponding electron density maxima with underestimated X-H (X = C, O or N) distances and a large ambiguity. The positional optimization was judged to be successful by the O-H and N-H bond lengths (1.03–1.06 Å, Supplementary Table 1). In particular, the O-H bond length versus the H···O distance (or N-H versus H···N) follow their relations available from the neutron diffraction data set of O-H···O (or N-H···N) bonded systems[29,30]. The reference hypothetical paraelectric structure ($\lambda = 0$) was constructed from this ferroelectric structure by adding inversion symmetry. Supplementary Table 2 summarizes the changes in the space- and point-group symmetries between the ferroelectric and hypothetical paraelectric structures. The electronic structures and corresponding spontaneous polarizations were calculated for different degrees of polar distortion $\lambda$ between the centric (hypothetical paraelectric, $\lambda = 0$) and fully polar (ferroelectric, $\lambda = 1$) configurations.

For each compound, the validity of the simulations was confirmed by the smooth $\lambda$-dependence of the polarization (Fig. 2) as well as the polarization vectors lying along the symmetrically allowed direction. The $\mathbf{P}^{cal}$ values of the CRCA and CBDC crystals are close to the corresponding theoretical predictions: $(P_a, P_b, P_c) = (0, 0, 26.0)$ $\mu$C cm$^{-2}$ for CRCA[12] and $(P_a, P_b, P_c^*) = (12.7, 0, -6.6)$ $\mu$C cm$^{-2}$ for CBDC[22]. These values change slightly depending on the details of the computational conditions such as the structural parameters and the exchange-correlation functional. For reference, Picozzi et al.[31] reported polarizations of 24–32 $\mu$C cm$^{-2}$ for CRCA using various exchange-correlation functionals.

In Table 1, we compare the theoretical polarizations $\mathbf{P}^{cal}$ with the $P - E$ hysteresis data $\mathbf{P}^{exp}$. For the PhMDA, MBI and DC-MBI crystals, the remanent polarizations listed are those of the previous report[16,17] and agree well with the corresponding theoretical polarizations. Since all these crystals were commonly grown from a high-temperature vapour phase, the thermal annealing effect is likely to have maximized the performance by eliminating pinned DWs. In fact, PFM images of MBI crystals revealed only neutral 180° and 90° DWs rather than charged DWs[17].

However, for all the solution-processed as-grown single crystals, the $\mathbf{P}^{cal}$ values (Table 1) are much larger than previously reported values[16]: CRCA ($P_c = 21$ $\mu$C cm$^{-2}$), HPLN (3.0 $\mu$C cm$^{-2}$, normal to the (10$\bar{1}$) plane), and CBDC (($P_a, P_c) = (0.9, 2.9)$ $\mu$C cm$^{-2}$). The crystals were grown from slow evaporation of solution at close to room temperature: CRCA from 1 N hydrochloric acid, $\alpha$-form HPLN from ethanol, and CBDC from water. These observations, indicative of the strong DW pinning, necessitated us to excite DW motion through a preconditioning treatment such as thermal annealing and/or repetitive switching.

**Optimization of solution-grown ferroelectrics**. All the previous measurements and optimizations of polarization switching of CRCA, HPLN, and CBDC crystals were conducted in air or inert gas. The present re-examinations have reinforced the maximum amplitude ($E_{max}$) of the bipolar electric field, which was hitherto set to less than 33 kV cm$^{-1}$ for a triangular waveform so as to avoid an electric discharge between electrodes on the crystal surface. Satisfactory optimization has been accomplished for crystals immersed in a silicone oil with field amplitudes increased up to 33–100 kV cm$^{-1}$ using rectangular-pulse voltages (Fig. 3).

Earlier work on a CRCA crystal increased $P_r$ up to 21 $\mu$C cm$^{-2}$ by applying 600 cycles of a 1 Hz triangular-wave voltage of

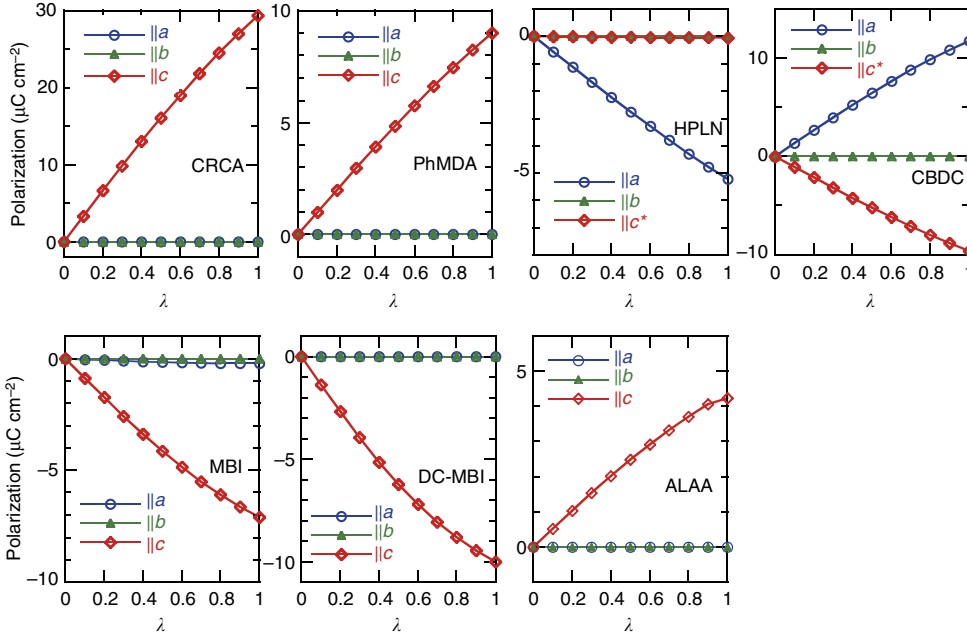

**Figure 2 | Theoretical calculations of polarization for seven PTM ferroelectrics.** Evolution of spontaneous polarization as a function of degree of polar distortion $\lambda$ changing from the centrosymmetric reference (hypothetical paraelectric, $\lambda = 0$) to the fully polarized (ferroelectric, $\lambda = 1$) configurations.

**Table 1 | Experimental and calculated polarization of PTM ferroelectric crystals.**

| Compound | Polarization ($\mu C\,cm^{-2}$) | | | | | | Direction component $(x, y, z)$ | Local dipole moment of proton transfer $|\mu|$ ($10^{-30}$ C m) |
|---|---|---|---|---|---|---|---|---|
| | Experimental | | Theoretical (Berry) | | Point charge model | | | |
| | $|P^{exp}|$ | $(P_x, P_y, P_z)^{exp}$ | $|P^{cal}|$ | $(P_x, P_y, P_z)^{cal}$ | $|P^{ion}|$ | $(P_x, P_y, P_z)^{ion}$ | | |
| **1.** CRCA | 30 | (0, 0, 30) | 29.4 | (0, 0, 29.4) | 5.5 | (0, 0, 5.5) | $(a, b, c)$ | 4.4, 5.0 |
| **2.** PhMDA | 9 | (0, 0, 9) | 9.0 | (0, 0, 9.0) | 1.8 | (0, 0, 1.8) | $(a, b, c)$ | 5.3 |
| **3.** HPLN | 5.6* | — | 5.2 | (−5.2, 0, −0.1) | 2.5 | (0, 0, 0) | $(a, b, c^*)$ | 5.9, 6.0 |
| | | 4.5 | | 4.2 | | | $\perp (10\bar{1})$ | |
| **4.** CBDC | 13.2 | (8.6, 0, −10.0)† | 15.1 | (11.7, 0, −9.6) | 6.7 | (6.3, 0, −2.2) | $(a, b, c^*)$ | 5.5 |
| **5.** MBI | 7.4* | — | 7.1 | (−0.2, 0, −7.1) | 2.5 | (−0.2, 0, −2.5) | $(a, b, c^*)$ | 6.7, 5.7, 6.7, 6.1 |
| | 5.2 | | 5.0 | | | | $\|[101]$ | |
| **6.** DC-MBI | 10 | (0, 0, −10) | 10.0 | (0, 0, −10.0) | 3.8 | (0, 0, 3.8) | $(a, b, c)$ | 8.3 |
| **7.** ALAA | 3.6 | (0, 0, 3.6) | 4.2 | (0, 0, 4.2) | 0.6 | (0, 0, 0.6) | $(a, b, c)$ | 7.5 |

ALAA, 3-anilinoacrolein anil; CBDC, cyclobutene-1,2-dicarboxylic acid; CRCA, croconic acid; DC-MBI, 5,6-dichloro-2-methylbenzimidazole; HPLN, 3-hydroxyphenalenone; PhMDA, 2-phenylmalondialdehyde; PTM, prototropy.
*The direction of the applied field exhibits a certain inclination angle from the polarization vector. The total amplitude of experimental polarization $|P^{exp}|$ was derived in consideration of this (theoretical) angle.
†Because the sign of each direction component of $P^{exp}$ could not be identified by the $P-E$ hysteresis experiments alone, it is assumed to be the same as that of $P^{cal}$.

$E^{max} = 33\,kV\,cm^{-1}$ followed with thermal annealing at 400 K (ref. 12). Here, we applied bipolar rectangular-pulse voltages of a stronger field amplitude (pulse field amplitude $E^{puls} = 33$–$55\,kV\,cm^{-1}$) to four specimens of different initial $P_r$ ranging from 2 to 13 $\mu C\,cm^{-2}$. Gradual expansion of the hysteresis loops with an increasing number of pulses corresponds to the so-called wake up process[32]. After a few ten thousand pulses, the $P_r$ reproducibly reached a maximum as high as 28–32 $\mu C\,cm^{-2}$ (Supplementary Fig. 9). This optimum $P_r$ is in excellent agreement with our theoretical evaluation (29.4 $\mu C\,cm^{-2}$) and also with the recent studies of Picozzi et al.[31] Moreover, the switchable polarization breaks its own record for organic ferroelectrics by increasing from 21 to about 30 $\mu C\,cm^{-2}$ and has just exceeded the performance of some commercial ferroelectric materials such as BaTiO₃ and SrBi₂Ta₂O₉

(20–26 $\mu C\,cm^{-2}$)[1,33]. For the crystal specimen exemplified in Fig. 3a, the optimized $P-E$ loops are improved in their rectangularity and frequency independence of $P_r$ in comparison with the corresponding earlier data. It should be noted that such strong polarization can be fully switched with a low coercive field (34 $kV\,cm^{-1}$) even at a frequency as high as 1 kHz.

Polarizations of the α-form HPLN crystal were improved with both thermal heating and repetitive switching (Supplementary Fig. 10). Although the calculated polarization vector is parallel to the a-axis, only well-developed crystal surfaces available for experiments were the $(10\bar{1})$ and $(\bar{1}01)$ planes (Supplementary Fig. 3c). Through thermal annealing, we noticed that the ferroelectric phase is thermally robust at least up to $T = 425\,K$, at which temperature the $P-E$ curves are quasi-rectangular with $P_r = 5.0\,\mu C\,cm^{-2}$ and $E_c \sim 10\,kV\,cm^{-1}$ at frequencies of

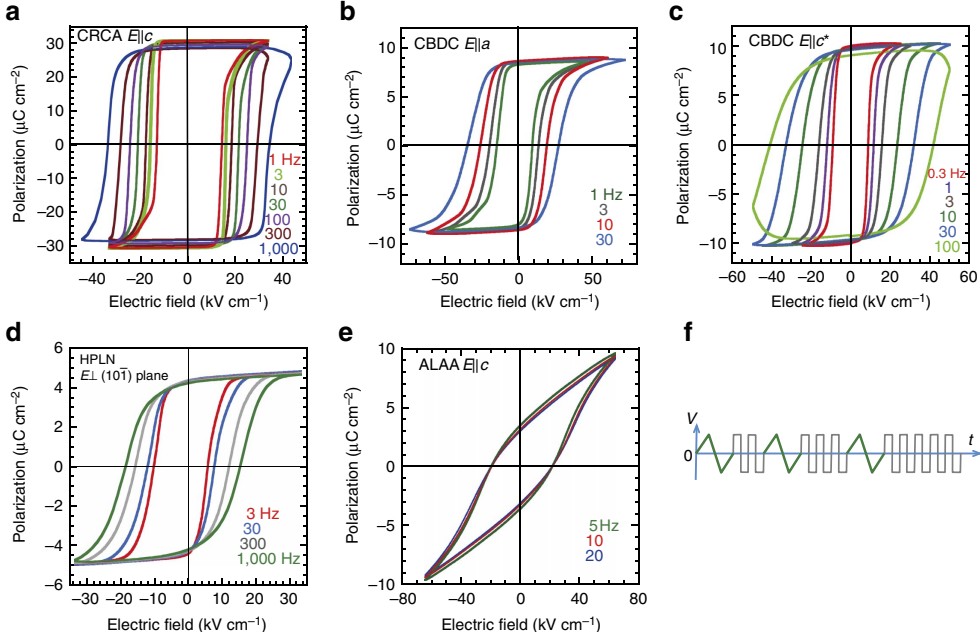

**Figure 3 | Polarization-optimized ferroelectric switching at room temperature.** Electric polarization ($P$) versus electric field ($E$) hysteresis loops measured with a triangular ac electric field of various frequencies for (**a**) $c$-direction polarization of CRCA, (**b,c**) $a$- and $c^*$-direction polarization of CBDC, (**d**) polarization normal to the ($10\bar{1}$) plane of HPLN, and (**e**) $c$-direction polarization of ALAA. (**f**) A schematic waveform used for optimization in an endurance measurement mode.

30–1,000 Hz (Supplementary Fig. 10c). The switchable polarization decreased slightly after cooling and additionally with time at room temperature, but it could be optimized to $P_r = 4.5\,\mu C\,cm^{-2}$ again by continuously applying bipolar rectangular-pulse voltages of a strong field ($33\,kV\,cm^{-1}$) at room temperature (Fig. 3d). Considering the inclination angle between $\mathbf{P}^{cal}$ ($\sim\|a$) and the applied field $\mathbf{E}$, the observed component of $\mathbf{P}^{exp}$ along $\mathbf{E}$ corresponds to a total amplitude $|\mathbf{P}^{exp}|$ of $\sim 5.6\,\mu C\,cm^{-2}$.

Substantial improvements have also been achieved in switching the CBDC crystal (Supplementary Fig. 11). The crystal polarity is uniaxial and lies within the crystallographic $ac$ plane by symmetry. Both the $a$- and $c^*$-direction polarizations were maximized by applying stronger-field cycles: bipolar triangular-waveform voltages ($E^{max} = 60\,kV\,cm^{-1}$) and rectangular-pulse voltages ($E^{puls} = 45\,kV\,cm^{-1}$), respectively (Fig. 3b,c). The optimized polarization $\mathbf{P}^{exp}$ exhibited $P_a = 8.6\,\mu C\,cm^{-2}$ and $P_c{}^* = 10.0\,\mu C\,cm^{-2}$, which did not depend on frequencies up to 30 Hz. This $\mathbf{P}^{exp}$ is comparable to the calculated $\mathbf{P}^{cal}$ in both magnitude (13.2 versus $15.2\,\mu C\,cm^{-2}$) and direction, as depicted by the open bold arrows in Supplementary Fig. 11c. It should be noted that these polarization vectors are almost parallel to the hydrogen-bonded chain running along the crystal [$\bar{2}01$] direction (solid arrow).

The ALAA crystal, which was newly grown from ethanol solution, also exhibited ferroelectricity in the $P-E$ hysteresis experiments with an $\mathbf{E}\|c$ configuration (Fig. 3e), in agreement with the structural assessment above. The remanent polarization was similarly optimized to a modest value ($3.6\,\mu C\,cm^{-2}$) after applying $3 \times 10^4$ cycles of triangular-waveform voltages ($E^{max} = 60\,kV\,cm^{-1}$). The ferroelectric state is stable at least up to the melting point at 388 K, below which the signature of the phase transition was absent until room temperature according to a thermal analysis using a high-sensitivity differential scanning calorimeter (Supplementary Fig. 12b).

For the solution-processed as-grown PTM ferroelectric crystals, the switchable polarization has been 'woken up' by repetitive

switching with an increased amplitude of the bipolar field rather than by improving the crystal quality. Hence, most of the pinning sites are not permanently clamping impurities or defects. Rather, they are charged DWs with compensating charges trapped nearby and can be gradually moved away under influence of a strong electric field. Note that the remanent polarizations become independent of the applied field frequency after optimization in all hysteresis loops (Fig. 3a–e). After the 'wake-up' process, polarization fatigue started with a steep increase in the coercive field $E_c$ beyond $\sim 10^5$ cycles for CRCA, CBDC, and ALAA and beyond $\sim 10^6$ cycles for HPLN. These 'wake-up' and fatigue behaviours are quite analogous to those of domain depinning and fatigue observed in some hard ferroelectric oxides[32,34–38]. In the latter ferroelectrics, the observed changes in the remanent polarization, coercive field, frequency dependence, and loop curvature have been well described by a model incorporating the variable interaction strength between the switchable dipole and fixed dipole as well as the depolarization field[35]. Similar arguments might be applied to the switching mechanism for ferroelectric PTM.

All the solution-processed as-grown PTM ferroelectric crystals herein has been finally optimized with significant improvement of their spontaneous polarization. In comparison with the corresponding earlier data, the improved performance is also evident by the optimized $P-E$ curvatures themselves (Fig. 3). Good rectangularity of the loop and frequency independence of $P_r$ suggested the almost complete removal of charged DWs and then successful disclosure of the genuine materials' properties.

**Comprehensive comparison of polarizations.** Figure 4a plots the best experimental performance $|\mathbf{P}^{exp}|$ of the seven PTM compounds as a function of the theoretical polarization amplitude $|\mathbf{P}^{cal}|$ along with that of anthranilic acid (ATA) form I[39], another hydrogen-bonded above-room-temperature ferroelectric. Because the applied field $\mathbf{E}$ is inclined with respect to the predicted $\mathbf{P}^{cal}$ for

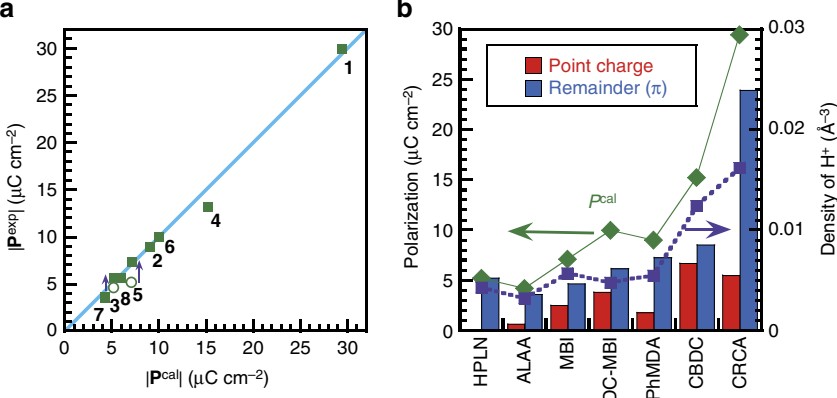

**Figure 4 | Comparison of spontaneous polarizations of PTM ferroelectrics.** (**a**) The experimentally optimized performance $|\mathbf{P}^{exp}|$ versus calculated total polarization $|\mathbf{P}^{cal}|$ of the CRCA (**1**), PhMDA (**2**), HPLN (**3**), CBDC (**4**), MBI (**5**), DC-MBI (**6**), and ALAA (**7**) crystals along with anthranilic acid I (**8**). The line indicates perfect agreement. For compounds **3** and **5**, open circles represent the raw data (**E**-direction component of $\mathbf{P}^{exp}$), whereas filled squares represent the corrected $|\mathbf{P}^{exp}|$ considering the inclination angle between $\mathbf{P}^{cal}$ and **E**. (**b**) Calculated total polarizations $|\mathbf{P}^{cal}|$ (green diamond) divided into ionic polarizations $|\mathbf{P}^{ion}|$ of displacing protons estimated under the point-charge approximation (red histogram) and the remainder contribution (blue histogram). Purple square denotes the volume density of protons in the crystal. Dotted and solid lines are guides for the eye.

the HPLN and MBI crystals, each measured $\mathbf{P}^{exp}$ was corrected to the total amplitude $|\mathbf{P}^{exp}|$ considering this inclination angle (arrows in the figure). Note that the polarizations vary widely from 3.6 to 30 μC cm$^{-2}$. After the optimization and correction, all the experimental data fall near the linear line $|\mathbf{P}^{exp}| = |\mathbf{P}^{cal}|$, and the largest discrepancy, found for the ALAA and CBDC crystals, is only 14–15%.

During the switching with PTM, the protons travel ∼0.6–1.0 Å within the hydrogen bond. This classical picture suggests significant ionic polarization. The line graph in Fig. 4b actually reveals a trend that shows the total polarization $|\mathbf{P}^{cal}|$ (solid diamonds) increasing with proton density (solid squares). We first estimate this electrostatic contribution from protons displacing among molecular (anionic) cores. In the point-charge model, the ionic polarization $\mathbf{P}^{ion}$ is expressed as

$$\mathbf{P}^{ion} = \sum_{(cell)} Z_i |e| \mathbf{u}_i / \Omega \tag{1}$$

where $e$ is the electron charge, $\mathbf{u}_i$ is the relative displacement of the static charges $Z_i |e|$, $\Omega$ is the unit cell volume, and $Z_i$ is taken as $+1$ for protons and $-1$ or $-2$ for molecular cores. The negative point charges were placed at the centre of gravity of the π-conjugated cores: the pentagon $C_5O_5{}^{2-}$ for CRCA, β-diketone enol $C_3O_2{}^-$ for PhMDA and HPLN, ethylenedicarboxylate $C_4O_4{}^{2-}$ for CBDC, imidazole ring $C_3N_2{}^-$ for MBI and DC-MBI, and bridging $C_3N_2{}^-$ unit for ALAA. The positive point charges were placed on protons at the energetically optimized locations calculated above. The $\mathbf{u}_i$ of each point charge is the displacement from the hypothetical paraelectric structure constructed by imposing pseudo-symmetry elements on the ferroelectric structure.

Table 1 lists the calculated $\mathbf{P}^{ion}$ and the magnitude of the local dipole moment $|\boldsymbol{\mu}|$ of the crystallographically independent protons. The dipole moment $\boldsymbol{\mu}_i$ was calculated from the proton displacements $\mathbf{u}_i$ by

$$\boldsymbol{\mu}_i = -|e| \mathbf{u}_i \tag{2}$$

under the condition that the centres of gravity of the anion displacements $\mathbf{u}_j$ were fixed in the cell: $\Sigma_{(cell)}\, \mathbf{u}_j = 0\ (j \in \text{anion})$. The N-H···N bonded compounds exhibited a slightly higher $|\boldsymbol{\mu}_i|$ ($5.5$–$8.3 \times 10^{-30}$ C m) than the O-H···O bonded compounds ($4.4$–$6.0 \times 10^{-30}$ C m) due to the longer separation of the two equilibrium proton positions. Because $\boldsymbol{\mu}_i$ reflects small variations

in the material, its density and direction are crucial factors determining the magnitude of $\mathbf{P}^{ion}$.

In the histogram in Fig. 4b, each total polarization $\mathbf{P}^{cal}$ was divided into $\mathbf{P}^{ion}$ (red bar) and a remainder contribution (blue bar). Each molecule of CRCA or CBDC accommodates two protons in the compact molecular size, and the resulting large proton density amplifies $\mathbf{P}^{ion}$. However, $\mathbf{P}^{ion}$ is still less than half of $\mathbf{P}^{cal}$ in amplitude, indicating the addition of some larger contributions. This is true also for the MBI, DC-MBI, and PhMDA crystals. It should be noted that the HPLN and ALAA crystals reveal a nearly zero $\mathbf{P}^{ion}$ due to the orthogonality of the proton's path and the chain dipole. Their large polarizations then manifest from the significant contribution of the remainder mechanisms. The most important contributions to $\mathbf{P}$ other than $\mathbf{P}^{ion}$ should come from the sections that experience the most dramatic redistribution of charge during the polarity reversal. The corresponding origin is nothing but switchable π-bond dipoles, the heart of PTM, which interchanges the locations of a single bond and adjacent double bond. Note that $\mathbf{P}^{exp}$ and $\mathbf{P}^{cal}$ of ALAA are smaller than those of HPLN despite very similar molecular size and switchable π-bond fragment. The reduced polarization could be explained by the significantly inclined orientation of switchable π-bond dipole from the bulk polarization vector (Supplementary Fig. S6).

**Comparison of switching field.** For hydrogen-bonded ferroelectrics such as KDP ($KH_2PO_4$) and its isomorphs, the physical properties are closely related to the local hydrogen-bond geometry, which critically affects the potential barrier height for proton hopping between two equilibrium positions[1,40,41]. Likewise, a series of ferroelectric supramolecules of anilic acids exhibited positive relationships in which stretching the hydrogen-bonded length enhanced both the phase-transition temperature and polarization performance[42]. Although the hydrogen bond lengths vary across the seven PTM ferroelectrics, they cannot be related to the thermal stability because of the lost paraelectric state[43]. As noted above, the bulk polarization strongly depends on whether the proton motion is nearly parallel or normal to the crystal polar axis. In turn, we found structural effects on the switching field. The $P-E$ hysteresis loops provide the coercive field $E_c$, which is determined by the field at the $P=0$ intercept and is accompanied by the peak (displacement) current in the

corresponding current-field ($I - E$) curve. The $E_c$ values measured at 10 Hz are plotted as a function of $O \cdots O$ or $N \cdots N$ distance in Fig. 5. The O-H$\cdots$O and N-H$\cdots$N bonded ferroelectrics similarly revealed respective relationships that indicate stretching the distance hardens the ferroelectric switching. This observation suggests that proton hopping would be a rate-limiting process during the switching, which occurs at an $E_c$ two orders of magnitude lower than those of polymers such as PVDF (around several hundred kilovolts per centimetre).

## Discussion

Our success in efficient optimization of ferroelectric switching has reduced the discrepancy of the polarization with the results of the first-principles calculations to less than 15%. Thus, one of the important outcomes of this work is the hallmark on the practicality of the calculations, which will be applied satisfactorily for the prediction of experimentally unknown performance of similar organic systems from the available precise crystal structure. It should be noted that evaluation of the ferroelectricity as well as the theoretical polarizations require careful diffraction studies. The unexpected finding of antiferroelectric DBT has demonstrated that routine structural assessment is not always straightforward. The other attractive findings are the strong optimized polarization and its low-field switching. Compared with ferroelectric polymers such as PVDF, the polarizations of the CRCA and CBDC crystals are stronger and those of the other compounds are comparable in magnitude. The CRCA crystal even breaks its own record for organic systems and also beats some commercial ferroelectric materials.

The PTM ferroelectrics studied herein exhibited a stable ferroelectric state up to temperatures well above room temperature even until the stability limit of the crystals themselves, such as the melting, decomposition, or sublimation temperatures. Two issues are relevant to this thermal stability. The first is the necessity of the depinning process for the solution-processed as-grown crystals. This is because the solution process, at far below the Curie point, spontaneously grows polarized crystals of a multidomain structure (that is, twinning) and embeds charged DWs frozen therein. The second is the excellent consistency between the experimental and simulated polarizations. One of the reasons for this is the deep potential minima in the fully polarized state, which minimized the ambiguity in the proton locations once accurate positions of the non-hydrogen atoms were experimentally determined.

The structure-property relationship together with the point-charge-model analysis yielded some molecular and crystal design strategies for higher-performance PTM ferroelectrics. The first issue concerns the direction of each contributing dipole moment. The total polarization $\mathbf{P}^{cal}$ was described by the accumulation of the ionic contribution $\mathbf{P}^{ion}$ from the proton displacement and the larger remaining contributions mainly from switchable π-bond dipoles. These cumulative contributions to the polarization are just converse to the subtractive nature in intramolecular hydrogen-bonded PTM such as 9-hydroxyphenalenone: the π-bond dipole changes in the opposite direction to that of the relocating protons, and some cancellation yields tiny switchable molecular dipoles ($0.4 \quad D \sim 1.3 \times 10^{-30} \, C \, m$)[44] compared with the local dipole moment around the hydrogen atom[45]. In contrast, the extended chains of intermolecular hydrogen bonds are ideal for enlarging the polarization, especially when both the local proton motion and the change in the π-bond dipoles are nearly aligned parallel to the polar crystal axis. This requirement is well satisfied in the CRCA, CBDC, and DC-MBI crystals.

The second issue is the density of dipoles. As noted above, the total polarization actually increases with increasing proton density. Similarly, one can easily envisage that the corresponding strategy for switchable π-bond dipoles is to increase the volume fraction of PTM fragments, which are specified in red in Fig. 1. As a typical case, the dibasic acid CRCA and CBDC are similar in terms of the ionic contribution $\mathbf{P}^{ion}$ and the proton density, but the remainder contribution in CRCA is huge and about three times as large as that of CBDC. The reason for this can be understood qualitatively in that the only fragment irrelevant to PTM is one $C = O$ unit in CRCA and much smaller than the cyclobutene $C_4H_4$ unit in CBDC. To summarize, the record-breaking performance of CRCA can be attributed to the ideal arrangement of dense dipoles; that is, the effective addition of ionic and π-bond polarizations, the highest spatial density of protons, and spatially dense PTM fragments extending across nearly the whole $C_5O_5$ core.

There are a number of useful tools established herein for further development of organic ferroelectric materials: the molecular design principles, structural assessment, theoretical simulation, and optimization procedures considering DW dynamics. Our results indicate that the optimization of molecular, crystal, electronic, and/or domain structures can bring more insight into the goals of higher performance, new functionalities, and useful applications of organic systems.

## Methods

**Sample preparation.** ALAA purchased from Alfa Aesar was recrystallized twice from slow evaporation of ethanol solution at 5 °C to afford new ferroelectric crystals of elongated orange plates. DBT was synthesized according to the literature[26], purified by vacuum sublimation in the temperature gradient, and crystallized from cold methanol. CRCA from Tokyo Chemical Industry was recrystallized three times from 1 N hydrochloric acid solution evaporated slowly under a stream of argon gas, and thereby collected as yellow plates. HPLN from Acros Organics was purified by a few repetitions of vacuum sublimation in the temperature gradient. CBDC purchased from Wako Pure Chemical Industries was recrystallized three times from acetonitrile. Orange plates of HPLN (α-form) and colourless plates of CBDC were grown under slow evaporation of the ethanol and aqueous solutions, respectively. For other compounds, we adopted almost the same purification and crystallization procedures as those in previous work.

**Crystallographic studies.** The X-ray diffraction data collection for the ALAA crystal at room temperature and the assignment of the crystallographic axes of the bulk single crystals were completed using a four-circle diffractometer equipped with a hybrid pixel detector (Rigaku AFC10 with PILATUS200K; graphite-monochromated MoKα radiation). The intensity data were analysed with the CrystalStructure crystallographic software packages (Molecular Structure Corp. and Rigaku Corp.). The final refinements were done with anisotropic atomic displacement parameters for the non-hydrogen atoms and with a fixed C-H bond length of 0.95 Å for the hydrogen atoms.

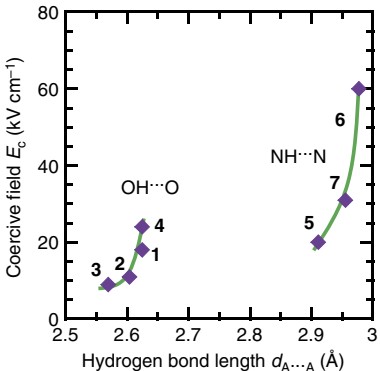

**Figure 5 | Coercive field as a function of hydrogen bond length.** The $O \cdots O$ distance is employed for the CRCA (**1**), PhMDA (**2**), HPLN (**3**) and CBDC (**4**) crystals, while the $N \cdots N$ distance is employed for the MBI (**5**), DC-MBI (**6**) and ALAA (**7**) crystals. Solid curves are guides for the eye. Each bond distance is obtained by averaging over crystallographically independent sites.

**Electric measurements.** All the electric measurements employed single crystals with painted gold paste as the electrodes. The $P - E$ hysteresis curves were measured by the virtual ground method[46] using a ferroelectrics-evaluation system (Toyo Corporation, FCE-1), which consists of a current/charge-voltage converter (Toyo Corporation Model 6252), arbitrary waveform generator (Biomation 2414B), analogue-to-digital converter (WaveBook 516), and voltage amplifier (NF Corporation, HVA4321). All the crystals were immersed in silicone oil to avoid electric discharge with a maximum electric field exceeding $30\,kV\,cm^{-1}$. The high-temperature measurements and/or annealing were also conducted by heating the sample in the oil bath.

**First-principles calculations.** First-principles computational code QMAS[47] based on the projector augmented-wave method[48] and the plane-wave basis set was employed for calculations of spontaneous polarization $\mathbf{P}^{cal}$ through the Berry phase formalism[49,50]. To describe the electronic exchange-correlation energy, the Perdew-Burke-Ernzerhof (PBE) version of the generalized gradient approximation (GGA) was used[51]. The target ferroelectric structures (degrees of polar distortion $\lambda = 1$) were constructed from the atomic coordinates of all the non-hydrogen atoms determined by the previous X-ray diffraction studies at room temperature. The locations of the hydrogen atoms were computationally relaxed so as to minimize the total energy. Refcodes of the corresponding CIF files, which we previously deposited in the CSD and used for calculations herein, are GUMMUW02 (CRCA), PROLON01 (PhMDA), TAPZIT01 (HPLN), CBUDCX01 (CBDC), KOWYEA (MBI) and REZBOP (DC-MBI).

**Data availability.** X-ray crystallographic data have been deposited with the Cambridge Crystallographic Data Centre (CCDC) under deposition numbers CCDC-1498921-1498923 and can be obtained free of charge from the Centre via its website (www.ccdc.cam.ac.uk/getstructures). All other data supporting the finding of this study are available within the article and its Supplementary Information.

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

## Acknowledgements

This work was partially supported by JSPS KAKENHI Grant Numbers JP16H02301 and JP26102014, and by CREST, Japan Science and Technology Agency (JST). The synchrotron X-ray study was performed with approval of the Photon Factory Program Advisory Committee (No. 2014S2-001). We thank Satoru Inoue for the synthesis of DBT.

## Author contributions

S.H. prepared the purified single crystals, performed the dielectric measurements, conceived the study design, and wrote most of the paper. S.I. performed the theoretical calculations. K.K. and R.K. contributed the diffraction experiments.

## Additional information

**Competing financial interests:** The authors declare no competing financial interests.

