## [Peer Review File · Nature Communications]

Reviewers' comments:

Reviewer #1 (Remarks to the Author):

I really liked this paper. It was sound, thorough and extremely well written (literally no typos!!). About the only comment I have for the authors is this: can they add the switching transients to Fig.5? this will throw some light on the dynamics of this family of ferroelectrics, in comparison to the perovskites which are typically limited by the velocity of sound.

This paper should be accepted.

Reviewer #2 (Remarks to the Author):

The present paper on organic ferroelectrics is a continuation of the authors' long and distinguished studies that began with croconic acid (CRCA, Fig. 1) a decade ago. So much history is a mixed blessing since advances have to be put in context. I have strongly recommended publication of several of the previous studies. I am less enthusiastic about the presentation here although the results are impressive. The paper reads like an executive summary of extensive results that are cited or found in more than 10 supplements.

The Discussion lists the advances in this paper. (1) First-principle calculations of the polarization P to accuracy of 15% with experiment; the cover picture, Table 1 and Fig. 4 make the point. (2) Thermal and field processing of solution grown materials that increases P by depinning or annealing charge domain walls (DM); such "awakening" is not needed for vapor grown material, as shown by direct imaging. (3) Improved P and switching for (processed) CRCA that now seems to be competitive with inorganic ferroelectrics. (4) The new (seventh) ferroelectric ALAA (Fig. 1). (5) The dipole density and the calculated fractional contributions to P of ions (H-bonds) and pi-systems.

As a theorist, I am impressed by the Berry phase calculation of P . The paragraph on First-Principle Calculations merely lists and cites the methods that go into Table 1. I doubt the "prediction of experimentally unknown performance" mentioned in the Discussion. An accurate structure is required (the ALAA structure from a data bank had to be redone) and is unlikely to precede a measurement of P . Perhaps I misunderstand, but higher P is assured for similar PTM if the active mole fraction is increased (densified).

It took a second reading to put the paper in context. The cooperative proton tautomerism mechanism (PTM) and the organic ferroelectrics mentioned in the Introduction cannot be appreciated even qualitatively without the nice summary in Fig. 1 at the beginning of Section II. I suggest moving Fig. 1 close to PTM. The acronyms in Fig. 1 could then be used in the historical comments on p. 3 without the full chemical names. Some readers will appreciate the names, others the PTM and still others the potential applications or DMs; only a very few (not me) will be conversant with all of them.

The new ferroelectric ALAA (fig. 1) has small P and orthogonal ionic/pi contributions as previously found in HPLN. Since DBT (p. 6) turns out to be an antiferroelectric, I wonder why it deserves a paragraph and several supplements.

The optimization of solution-grown ferroelectrics by a combination of thermal annealing and voltage pulses is interesting and will surely be written up more completely elsewhere. But the focus here is on the improved P-E curves in Fig. 4 and the material properties modeled in Table 1 rather than on how to get rid of charged DMs.

Two small points: (1) Define polyvinylidene fluoride (PVDF, $(\text{CH}_2\text{CF}_2)_n$) on p. 2; the chemical formula

is more easily recognized than an acronym or a name – PVDF usually stands for polyvinylidene difluoride. (2) At the beginning should “high Curie point (working temperature)” be (above the working temperature)? I am not familiar with the convention.

I think that revisions are needed. The new results presented are interesting and important but are to some extent also extensions of previous work. The threshold here is a matter of opinion. So is my view that the main points should be emphasized at the expense of other studies and properties of organic ferroelectrics.

Author's response to the Reviewer #1

Comment:

Reviewer #1 (Remarks to the Author):

I really liked this paper. It was sound, thorough and extremely well written (literally no typos!!). About the only comment I have for the authors is this: can they add the switching transients to Fig.5? this will throw some light on the dynamics of this family of ferroelectrics, in comparison to the perovskites which are typically limited by the velocity of sound..

Reply:

We appreciate so much the reviewer's kind and encouraging comments. Unfortunately, we have not measured time-evolution of switching current, so we cannot add the corresponding information.

Author's response to the Reviewer #2

Comment:

Reviewer #2 (Remarks to the Author):

The present paper on organic ferroelectrics is a continuation of the authors' long and distinguished studies that began with croconic acid (CRCA, Fig. 1) a decade ago. So much history is a mixed blessing since advances have to be put in context. I have strongly recommended publication of several of the previous studies. I am less enthusiastic about the presentation here although the results are impressive. The paper reads like an executive summary of extensive results that are cited or found in more than 10 supplements.

The Discussion lists the advances in this paper. (1) First-principle calculations of the polarization P to accuracy of 15% with experiment; the cover picture, Table 1 and Fig. 4 make the point. (2) Thermal and field processing of solution grown materials that increases P by depinning or annealing charge domain walls (DM); such "awakening" is not needed for vapor grown material, as shown by direct imaging. (3) Improved P and switching for (processed) CRCA that now seems to be competitive with inorganic ferroelectrics. (4) The new (seventh) ferroelectric ALAA (Fig. 1). (5) The dipole density and the calculated fractional contributions to P of ions (H-bonds) and pi-systems.

As a theorist, I am impressed by the Berry phase calculation of P . The paragraph on First-Principle Calculations merely lists and cites the methods that go into Table 1. I doubt the "prediction of experimentally unknown performance" mentioned in the Discussion. An accurate structure is required (the ALAA structure from a data bank had to be redone) and is unlikely to precede a measurement of P . Perhaps I misunderstand, but higher P is assured for similar PTM if the active mole fraction is increased (densified).

Reply:

We thank the reviewer's encouraging comment on this work as impressive results as well as many sharp comments and kind advices for improvement of the presentation.

In the third paragraph of the reviewer's comment, I agree with the necessity of precise knowledge of crystal structure prior to prediction of P . This point was lost in the previous manuscript. The revision was made in the second sentence of Discussion by adding "from the

available precise crystal structure”.

Concerning the analysis of ALAA, both the theoretical and experimental polarizations are smaller than those of HPLN having the similar orthogonal ionic/pi contribution. I agree with the comment that higher P is assured for similar PTM if the active mole fraction is increased (densified). In the case of ALAA, active mole fraction is very similar to that of HPLN, but bulk polarization is reduced due to the significant inclination of pi-bond dipole from the polar c-axis as can be seen in Figure S6. The corresponding argument is added in the last sentence of section “Comprehensive comparison of polarizations” : “Note that \mathbf{P}^{exp} and \mathbf{P}^{cal} of ALAA are smaller than those of HPLN despite very similar molecular size and switchable π -bond fragment. The reduced polarization could be explained by the significantly inclined orientation of switchable π -bond dipole from the bulk polarization vector (see Figure S6).”

Comment:

It took a second reading to put the paper in context. The cooperative proton tautomerism mechanism (PTM) and the organic ferroelectrics mentioned in the Introduction cannot be appreciated even qualitatively without the nice summary in Fig. 1 at the beginning of Section II. I suggest moving Fig. 1 close to PTM. The acronyms in Fig. 1 could then be used in the historical comments on p. 3 without the full chemical names. Some readers will appreciate the names, others the PTM and still others the potential applications or DMs; only a very few (not me) will be conversant with all of them.

Reply:

We have added “(Figure 1)” in the first sentence of third paragraph in Introduction. We are sorry for trouble by careless mistake that this reference was lost.

Comment:

The new ferroelectric ALAA (fig. 1) has small P and orthogonal ionic/pi contributions as previously found in HPLN. Since DBT (p. 6) turns out to be an antiferroelectric, I wonder why it deserves a paragraph and several supplements.

Reply:

The reviewer’s comment concerns that the new ferroelectric ALAA is relatively less focused than the antiferroelectric DBT. We believe that this unbalanced focus is somewhat improved by adding the discussion on the polarization of ALAA in comparison with HPLN, as replied above. The inclusion of unexpected antiferroelectricity in this study is intended not only to avoid the widespread of erroneous prediction but also to demonstrate that routine structural assessment is not always straightforward. The case indicates again the necessity of careful diffraction studies prior to evaluation of the ferroelectricity as well as the theoretical polarizations. This argument is added as the third and fourth sentences in Discussion section. We believe that this revision answers why DBT deserves to be discussed together with genuine ferroelectric PTM.

Comment:

The optimization of solution-grown ferroelectrics by a combination of thermal annealing and voltage pulses is interesting and will surely be written up more completely elsewhere. But the focus here is on the improved P-E curves in Fig. 4 and the material properties modeled in Table 1 rather than on how to get rid of charged DMs.

Reply:

We thank the sharp comment. This comment corresponds to the insufficient arguments in the section “Optimization of solution-grown ferroelectrics”. We have been aware that this section does not include proper message on important outcome that complete removal of charged DWs

led to the successful disclosure of the genuine materials' properties. In this revision, we added three sentences in the last of this section to emphasize this point. We believe that this revision also improves the connection with the subsequent section.

Comment:

Two small points: (1) Define polyvinylidene fluoride (PVDF, $(\text{CH}_2\text{CF}_2)_n$) on p. 2; the chemical formula is more easily recognized than an acronym or a name; PVDF usually stands for polyvinylidene difluoride. (2) At the beginning should "high Curie point (working temperature)"; be (above the working temperature)? I am not familiar with the convention.

Reply:

(1) The chemical formula has been added after the abbreviation "PVDF". (2) Whereas the Curie point is upper temperature limit of memory functions, it is not exactly the working temperature. To avoid the confusion, we have removed "working temperature".

Comment:

I think that revisions are needed. The new results presented are interesting and important but are to some extent also extensions of previous work. The threshold here is a matter of opinion. So is my view that the main points should be emphasized at the expense of other studies and properties of organic ferroelectrics.

Reply:

Thanks to the referee's comments and kind advices, we believe that the presentation is much improved and becomes satisfactory after this revision.

REVIEWERS' COMMENTS:

Reviewer #1 (Remarks to the Author):

I am happy with this revised manuscript

Reviewer #2 (Remarks to the Author):

The authors made small but pertinent revisions in response to my review. I recommend publication. The cumulative results are impressive and agreement has now been achieved between the measured and calculated P. The clear presentation that summarizes the state of the art in organic ferroelectrics is more accessible. Interested readers are referred to extensive supplementary material.

Line 63: typo: (RABH)

construct extended chains of intermolecular resonance-assisted hydrogen bonds (RHAB)¹⁸,

Author's response to the Reviewer

Comment:

Reviewer #1 (Remarks to the Author):

I am happy with this revised manuscript

Reply:

Thank you for kind review again.

Comment:

Reviewer #2 (Remarks to the Author):

The authors made small but pertinent revisions in response to my review. I recommend publication. The cumulative results are impressive and agreement has now been achieved between the measured and calculated P. The clear presentation that summarizes the state of the art in organic ferroelectrics is more accessible. Interested readers are referred to extensive supplementary material.

Line 63: typo: (RABH)

construct extended chains of intermolecular resonance-assisted hydrogen bonds (RHAB)18,

Reply:

Thank you for kind review again. The misspelling has been corrected.